# Timing Differences in Stride Cycle Phases in Retired Racehorses Ridden in Rising and Two-Point Seat Positions at Trot on Turf, Artificial and Tarmac Surfaces

**DOI:** 10.3390/ani13162563

**Published:** 2023-08-09

**Authors:** Kate Horan, Haydn Price, Peter Day, Russell Mackechnie-Guire, Thilo Pfau

**Affiliations:** 1Department of Clinical Science and Services, The Royal Veterinary College, Hawkshead Lane, Brookmans Park AL9 7TA, Hertfordshire, UK; pday@rvc.ac.uk; 2Little Pastures, Gwehelog, Usk NP15 1RD, Gwent, UK; haydnprice1@me.com; 3Equine Department, Hartpury University, Gloucester GL19 3BE, Gloucestershire, UK; russell.mackechnie-guire@hartpury.ac.uk; 4Faculties of Kinesiology and Veterinary Medicine, University of Calgary, 2500 University Dr NW, Calgary, AB T2N 1N4, Canada; thilo.pfau@ucalgary.ca

**Keywords:** equine, hoof, kinematics, trot, surfaces, jockey position

## Abstract

**Simple Summary:**

Racehorses routinely trot over tarmac, artificial and turf surfaces to access gallop tracks and during warm-up exercises. While undertaking these activities, jockeys may assume either a rising or two-point seat position. Understanding how hoof movements vary depending on jockey seating style may have a bearing on safety and stability, and this may vary across surfaces with contrasting fundamental properties, such as hardness and regularity. This study fitted inertial measurement units (IMUs) to the forelimb hooves of six retired Thoroughbred racehorses as they trotted in a randomized order over tarmac, artificial and turf surfaces, with their jockey in rising and two-point seat positions. The IMUs enabled hoof landing, mid-stance, breakover, and swing durations to be calculated, in addition to stride length, for each trial condition. Landing duration was significantly shorter on the tarmac than on the turf and artificial surfaces. Mid-stance duration was significantly longer on the tarmac than on the artificial surface and increased for the two-point seat position. Neither surface nor jockey position affected breakover, but the presence of a jockey increased breakover compared to in-hand exercise. Swing duration was significantly longer on turf compared to the artificial surface. Stride length was significantly shorter on tarmac than on turf, and stride length had a strong positive correlation with speed.

**Abstract:**

Injuries to racehorses and their jockeys are not limited to the racetrack and high-speed work. To optimise racehorse-jockey dyads’ health, well-being, and safety, it is important to understand their kinematics under the various exercise conditions they are exposed to. This includes trot work on roads, turf and artificial surfaces when accessing gallop tracks and warming up. This study quantified the forelimb hoof kinematics of racehorses trotting over tarmac, turf and artificial surfaces as their jockey adopted rising and two-point seat positions. A convenience sample of six horses was recruited from the British Racing School, Newmarket, and the horses were all ridden by the same jockey. Inertial measurement units (HoofBeat) were secured to the forelimb hooves of the horses and enabled landing, mid-stance, breakover, swing and stride durations, plus stride length, to be quantified via an in-built algorithm. Data were collected at a frequency of 1140 Hz. Linear Mixed Models were used to test for significant differences in the timing of these stride phases and stride length amongst the different surface and jockey positions. Speed was included as a covariate. Significance was set at *p* < 0.05. Hoof landing and mid-stance durations were negatively correlated, with approximately a 0.5 ms decrease in mid-stance duration for every 1 ms increase in landing duration (r^2^ = 0.5, *p* < 0.001). Hoof landing duration was significantly affected by surface (*p* < 0.001) and an interaction between jockey position and surface (*p* = 0.035). Landing duration was approximately 4.4 times shorter on tarmac compared to grass and artificial surfaces. Mid-stance duration was significantly affected by jockey position (*p* < 0.001) and surface (*p* = 0.001), speed (*p* < 0.001) and jockey position*speed (*p* < 0.001). Mean values for mid-stance increased by 13 ms with the jockey in the two-point seat position, and mid-stance was 19 ms longer on the tarmac than on the artificial surface. There was no significant difference in the breakover duration amongst surfaces or jockey positions (*p* ≥ 0.076) for the ridden dataset. However, the mean breakover duration on tarmac in the presence of a rider decreased by 21 ms compared to the in-hand dataset. Swing was significantly affected by surface (*p* = 0.039) and speed (*p* = 0.001), with a mean swing phase 20 ms longer on turf than on the artificial surface. Total stride duration was affected by surface only (*p* = 0.011). Tarmac was associated with a mean stride time that was significantly reduced, by 49 ms, compared to the turf, and this effect may be related to the shorter landing times on turf. Mean stride length was 14 cm shorter on tarmac than on grass, and stride length showed a strong positive correlation with speed, with a 71 cm increase in stride length for every 1 m s^−1^ increase in speed (r^2^ = 0.8, *p* < 0.001). In summary, this study demonstrated that the durations of the different stride cycle phases and stride length can be sensitive to surface type and jockey riding position. Further work is required to establish links between altered stride time variables and the risk of musculoskeletal injury.

## 1. Introduction

The daily routine of racehorses in training typically involves a commute at walk and trot across tarmac, dirt tracks, artificial surfaces and/or turf to access gallop tracks. A subsequent warm-up will involve similar exercise over a subset of these surfaces before high-speed work on either an artificial surface (mostly used in training in the UK) or turf (mostly used for races in the UK). During the commute and warm-up period, jockeys will opt to ride in either a rising or two-point seat position when the horse trots. Sitting trot is rarely used by jockeys and instead tends to communicate to racehorses that a trot-canter transition is required. The influence of rising versus two-point seat riding positions on racehorses’ biomechanics and the likelihood of horse or jockey injury is unknown. However, reports of racehorse and jockey injuries on access routes to gallops are high. For example, the tarmac roads in Middleham, Yorkshire, are a hotspot for racehorse and jockey injuries, and over 500 racehorses are ridden daily along the roads here to get to and from training gallops. Such hard surfaces, designed primarily for other road users, are deemed unfavourable compared to short, firm, well-drained turf, vegetated paths on a firm base, or non-slip surfaces (British Horse Society, 2021). The British Horse Society reported 15 mild–moderate equestrian road accidents in Middleham from July 2018 to July 2020, which occurred throughout all seasons (BHS, pers. comm.). Further, at least 14 incidents involving horses slipping and incurring muscle strain and/or horses falling were reported by riders to the local council between 6 November 2019 and 1 July 2020 (Byford, pers. comm.). These issues are frequently reported in the media [1,2]. It is, therefore, vital to understand how the safety and stability of racehorses and their jockeys can be optimized when travelling over varied terrains at trot to access gallop tracks. A better appreciation of how horses’ hoof and upper body kinematics might respond to jockey positions on different surfaces may be relevant for lessening these incidents.

In addition to acute injuries, understanding Thoroughbreds’ hoof and upper body kinematics on different surfaces is also relevant for their long-term health. For example, epidemiological studies have indicated that surface type may be associated with lameness in dressage, show jumping and racehorses [3,4,5,6,7]. In particular, firm surfaces have been associated with an increased risk of injury in galloping racehorses [3,4,8,9,10], and this may be related to the increased frequency and power of hoof accelerations on firmer surfaces [11,12,13]. Outputs from kinematic studies also suggest hoof, limb and upper body kinematics change in response to surface type [12,14,15,16,17,18]. For example, grass is associated with more upper body movement asymmetry in trotting horses compared to tarmac and synthetic surfaces, possibly due to high surface irregularity [12]. In addition, surface type can influence the centre of mass displacements of horses and their jockeys during galloping [18]. The degree of hoof slip post-landing also correlates with surface type [19], and a balance between acquiring sufficient slip on a particular surface to dissipate impact accelerations and forces versus a limit to prevent injury is required [20,21,22,23]. Hoof breakover is also a relevant consideration for efficient locomotion, as breakover occurs during the propulsive phase of the stride. The potential for injury may be increased during breakover because the risk of hoof slip may increase due to a reduced friction force as the limb unloads; in addition, during breakover, the point of zero moment moves from a more centred position towards the toe area resulting in an increased moment arm of the ground reaction force. Data from galloping horses indicate that breakover duration is reduced on an artificial surface compared to a turf surface [17], but if and how this, in turn, influences the degree of temporal coordination between horses and their jockeys has not previously been studied, either at gallop or in other gaits. Nonetheless, if ground surface type is a significant risk factor for racehorse injuries [9,10,24,25], this implies that surface conditions may impact horse-rider biomechanical stability and efficiency.

Furthermore, there is a lack of data regarding horses’ hoof kinematics on different surfaces in relation to the posture and actions of a rider, who has the potential to fundamentally alter a horse’s way of going. It is unclear how adjustments to rider position are reflected at the level of hoof-surface interactions, which are at the heart of slippage and falls. Jockey injuries are often linked to horse falls [26], which may be particularly catastrophic if, for example, horse slipping incidents coincide with passing vehicular traffic. This study aimed to investigate whether the ridden position of jockeys influences the hoof kinematics of trotting horses over three surface types (artificial, turf, tarmac) at the trot, with the jockey adopting two contrasting ridden positions, to facilitate a comparison of ridden states with different degrees of horse-rider coupling. The surfaces selected were expected to initiate fundamental differences in hoof kinematics, such as slip distance and ‘stride to stride’ variation: tarmac (hard and consistent), turf (variably hard and inconsistent), artificial (soft and intermediate consistency). We hypothesised that hoof landing times would be shortest on the firmer tarmac surface due to a reduced slip phase, and, as a result, we expected stride times to be reduced on this surface. We expected the timings of the stance and swing phases to be least consistent with the jockey in a two-point seat position, particularly when this less stable jockey position [27] is combined with a more irregular surface. We hypothesised that breakover would be faster with the jockey in the two-point seat, as the propulsion phase should occur with the additional mass of the jockey positioned further cranially and, as such, the horse should expend less time and energy accelerating the jockey’s mass forwards. We also predicted that this effect would be accentuated on the springy and deformable artificial surface, which should absorb a higher proportion of energy at hoof impact and return more energy to the hoof post-impact relative to the other surfaces [13]. This study is complemented by related research assessing the upper body kinematics of the horses and jockeys in the same ridden trials (Horan et al., in prep.). 

## 2. Materials and Methods

### 2.1. Ethics

Ethical approval for this study was received from the RVC Clinical Research Ethical Review Board (URN 2020 2001-2), and the participating jockey and horse owners provided informed consent.

### 2.2. Horse and Rider Participants

A convenience sample of six retired Thoroughbred racehorses in regular work and utilised for jockey training at the British Racing School (BRS) in Newmarket, UK, were included in this study. These horses are no longer involved in competitive racing but are fit and work under conditions similar to those encountered during active race training. The horses had ages between 7 and 19 years, their masses ranged from 510 to 580 kg, and their heights ranged from 16 to 17 hh (1.63 to 1.73 m). For all horses, the forelimb hooves were shod with steel shoes, and the hindlimb hooves were unshod. Before data collection, the horses’ gait asymmetries were evaluated during trotting in-hand on tarmac, and this information is available in the Appendix A. The same jockey rode all horses and had a mass of approximately 60 kg. The jockey has a category A and point-to-point license and approximately nine years of experience in the racing industry, typically riding five horses per day.

### 2.3. Equipment

Inertial measurement unit (IMU) devices (HoofBeat, Tolbert, The Netherlands) were fitted to the dorsal aspect of the horses’ front hooves using double-sided tape and Velcro. The sensors were purchased only a few months before the study and were under warranty. These devices are programmed to use an algorithm that generates information on median stride length, landing duration, stance duration, breakover duration and swing duration via an assessment of changes in hoof orientation. Their recording frequency is 1140 Hz. A preliminary validation study at walk and trot using 15 steps of data from one horse has indicated good agreement with an optoelectronic technique [28]. The sensors define landing as the time from initial contact until the hoof comes to a complete stop, and stabilisation of the hoof occurs with respect to the ground surface; mid-stance is the time from when the hoof has come to a complete stop on the ground until the heels start lifting; breakover is the time from when the heel buttresses come 5 mm off the ground until the last contact of the toe with the ground; and swing duration describes the time the hoof is not in contact with the ground [28,29]. The sensors additionally quantify speed.

### 2.4. Trial Conditions

The horse-jockey dyads performed ridden trials in both rising trot and two-point seat positions on tarmac, artificial and turf surfaces. For the rising condition, there were trials with the jockey sitting to both the left (defined as rising trot—left diagonal) and right (rising trot—right diagonal) forelimb stance phases. However, for this analysis, left and right diagonal data were not differentiated. The order of trials was randomised in case of carry-over effects, such as tiredness of the horse or jockey. However, as all horses were used to being exercised at all four paces as part of their daily involvement in jockey education, they were fit and capable of completing all trotting exercises required without becoming fatigued. Similarly, since the jockey selected for this study was used to riding racehorses daily, they were also fit to participate in multiple trials without feeling tired. Nevertheless, there were short rest breaks between each trial condition, during which equipment was moved if necessary (e.g., cameras for filming). The three surface types were adjacent, so the horses did not need to travel far between trials. The total time spent undertaking trotting exercises was around 20 min per horse.

During data collection, the artificial surface (Martin Collins Activ Track) was deemed to have ‘standard’ going, and the turf was ‘good-firm’. The grass was well-drained owing to the underlying chalk lithology. Each horse was measured over the same area for each surface type. The grass and artificial surfaces were not harrowed or rolled between trials, but visual inspection confirmed they maintained a good consistency throughout the data collection period, and weather conditions did not vary throughout the day. For reference, horses were trotted in-hand in a straight line over the tarmac wearing a bridle, with the handler on the horse’s left side. A conscious effort was made to ensure the horses’ heads and necks were unrestrained so head and neck positions did not bias data. The correct use of rising and two-point seat positions by the jockey was judged and confirmed by RMG, who is a BHSI equestrian coach, and HP, who was the lead consultant farrier for 20 years to the British Equestrian Federation (BEF) and World Class Equestrian Programme (WCP), and currently the consultant farrier to the Hong Kong Jockey Club performance programme.

### 2.5. Statistics

Linear mixed models were implemented using SPSS software (version 29.0.0) to assess the impact of surface and jockey position on stride length and median durations over stride cycles of hoof landing, stance, breakover and swing. Surface, jockey position, speed and surface*jockey position, surface*speed, and jockey position*speed interactions were defined as fixed factors. Horse ID was included as a random factor. Speed was included as a fixed covariate to account for variability in trot speed amongst trials and horses. The in-hand data, available only for the tarmac surface, was not included in the models. The *p* value outputs for the interaction terms of these initial linear mixed models were evaluated. If any *p* values for interaction terms exceeded 0.1, then these terms were removed so ‘final’ models could be run with fewer fixed terms to lower statistical noise. In each case, histograms of models’ residuals were plotted and inspected for normality. The significance threshold in all statistical tests was set at *p* < 0.05.

To eliminate extreme outliers from the ridden data sets before running the linear mixed models, we calculated boundaries for values 1.5 times the inter-quartile range below quartile 1 or 1.5 times the inter-quartile range above quartile 3 [30]. Data were deemed to be outliers if they fell outside these boundaries in full ridden datasets for landing duration, mid-stance duration, breakover duration, swing duration, speed, or stride length (i.e., data were not further sub-divided by surface or jockey position to identify outliers here).

## 3. Results

Available hoof kinematic data are summarised in Table 1. Please note that the hoof sensors occasionally experienced a signal failure resulting in incomplete data sets for each horse. However, linear mixed models are a robust statistical method capable of handling incomplete datasets, and the results from these models are detailed below (Table 2, Table 3, Table 4 and Table 5).

### 3.1. Landing Duration

Two outliers in the landing duration data were removed before running the Linear Mixed Model analysis. The preliminary linear mixed model indicated that the jockey position*speed and surface*speed interaction terms had high *p* values; *p* = 0.325 and *p* = 0.808, respectively. In the final model, with jockey position, surface, speed, and jockey position*surface included as fixed factors, surface (*p* < 0.001) and jockey position*surface (*p* < 0.035) were found to have a significant effect on median landing duration. Jockey position (*p* = 0.636) and trot speed had no effect (*p* = 0.679) (Table 2). Amongst the surface effects, the median landing duration was considerably reduced on the tarmac; it was 4.5 and 4.3 times lower than on the artificial and grass surfaces (Table 4). Pairwise comparisons, with Bonferroni correction, indicated that although the tarmac surface was significantly different from the other surfaces (*p* < 0.001), the grass and artificial surfaces were not significantly different.

When the surface and jockey position were considered together (Figure 1, Table 5), an interaction between these factors became apparent. On the grass surface, median landing duration increased slightly when the jockey adopted the two-point seat position, compared to the rising trot position. The same pattern was apparent for the tarmac surface, although this effect was proportionally larger, with an approximately 33% increase in landing duration for the two-point position. In contrast, on the artificial surface, the rising trot position appeared to be associated with a longer landing duration. A post-hoc analysis was used to confirm which surface+jockey position combinations were significantly different to one another. The pairwise comparisons are available in Appendix A. There were 8 out of 15 possible comparisons that were significant. Comparisons involving both grass and artificial surfaces were insignificant. The most contrasting conditions were rising trot on tarmac compared to rising on artificial (Δ66.2 ± 4.9 ms), followed by two-point on tarmac compared to rising on artificial (Δ61.9 ± 5.1 ms).

### 3.2. Mid-Stance Duration

Two outliers in the mid-stance duration data were removed before running the Linear Mixed Model analysis. The preliminary model indicated that the surface*speed interaction term had a high *p* value (*p* = 0.164). In the final model, with jockey position, surface, speed, and jockey position*surface and jockey position*speed interaction terms included as fixed factors, jockey position (*p* < 0.001), surface (*p* = 0.001), speed (*p* < 0.001) and jockey position*speed (*p* = 0.001) were all found to have significant effects on mid-stance duration. Jockey position*surface (*p* = 0.258) had no significant impact (Table 2). Mid-stance was significantly longer, by 13 ms, for the two-point seat position compared to the rising trot position (Table 3). Mid-stance duration decreased from tarmac to grass to artificial. However, only the tarmac and artificial surface were significantly different to one another (*p* < 0.001); they differed by 19 ms (Table 4). Figure 2B illustrates the relationship between mid-stance and speed. A weak positive relationship is defined by the equation y = 328 − 35 x, where y is mid-stance duration in ms and x is trotting speed in m s^−1^; r^2^ = 0.357, *p* < 0.001.

### 3.3. Breakover Duration

Three outliers in the breakover duration data were removed before running the Linear Mixed Model analysis (however, please note that these three values were not excluded from Figure 1 because they were not classified as outliers when the in-hand data was also considered). The preliminary model indicated that all interaction terms had high *p* values (*p* ≥ 0.478). In the final model, with jockey position, surface and speed included as fixed factors, no factors were identified to significantly affect median breakover duration (all *p* ≥ 0.076) (Table 2). However, although the in-hand data were not included in the main statistical models, it was apparent that breakover durations were considerably longer when the horses trotted in-hand on the tarmac surface (Figure 1). For this reason, an additional *t*-test (two sample, assuming equal variance) was used to compare the means for breakover in the ridden and in-hand data sets. The mean duration for breakover in the in-hand data was 65.3 ms (*n* = 10), whereas breakover had a mean duration of 45.6 ms (*n* = 18) for the ridden dataset, and the *t*-test revealed this difference was significant (*p* < 0.001).

### 3.4. Swing Duration

One outlier was removed in the swing duration data before running the Linear Mixed Model analysis. The preliminary model indicated that all interaction terms had high *p* values (*p* ≥ 0.302). In the final model, with jockey position, surface and speed included as fixed factors, surface (*p* = 0.039) and trot speed (*p* = 0.001) were found to have a significant effect on the median duration of the swing phase, but the jockey position was insignificant (*p* = 0.268) (Table 2). Amongst the surfaces, the grass had a significantly longer swing duration compared to the artificial surface, by 19.6 ms (*p* = 0.036), but other comparisons were not significantly different (*p* ≥ 0.176) (Table 4). A linear regression analysis demonstrated that swing duration (y in ms) showed a weak positive correlation with trot speed (x in m s^−1^) (Figure 2D: y = 275 +30 x, r^2^ = 0. 145, *p* = 0.008).

### 3.5. Stride Duration

One outlier from the entire stride duration data was removed before running the Linear Mixed Model analysis. The preliminary model indicated that all interaction terms had high *p* values (*p* ≥ 0.371). In the final model, with jockey position, surface and speed included as fixed factors, only surface was found to have a significant effect on median stride duration (*p* = 0.011); all other *p* values were ≥0.351 (Table 2). The tarmac surface was associated with a 49 ms shorter stride duration than grass (*p* < 0.010) (Table 4).

### 3.6. Relative Stance Time

The preliminary model indicated that all interaction terms had high *p* values (*p* ≥ 0.511). In the final model, with jockey position, surface and speed included as fixed factors, surface (*p* < 0.001) and speed (*p* < 0.001) were found to have significant effects on relative stance time but the jockey position had no significant effect (*p* = 0.688) (Table 2). Relative stance time was significantly lower on the tarmac surface compared to the grass (*p* = 0.007) and the artificial (*p* < 0.001) surfaces. A linear regression analysis indicated that relative stance time showed a weak negative correlation with trot speed: y = −3.5x + 56.7 when y is the stance time in ms and x is the speed in m s^−1^; r^2^ = 0.194.

### 3.7. Stride Length

The preliminary linear mixed model indicated that the jockey position*surface, jockey position*speed and surface*speed interaction terms had high *p* values; *p* = 0.815, *p* = 0.439 and *p* = 0.611, respectively. In the final model, with jockey position, surface and speed included as fixed factors, surface (*p* < 0.036) and speed (*p* < 0.001) were found to have a significant effect on stride length. Jockey position had no effect (*p* = 0.630) (Table 2). Estimated marginal mean stride lengths on the tarmac surface were 14 cm lower than on grass (*p* = 0.047). There was a strong positive correlation between stride length and speed (y= 70.9x − 7.67 when y is the stride length in cm and x is the speed in m s^−1^; r^2^ = 0.825, *p* < 0.001, Figure 2F)

### 3.8. Speed

The speed data per individual trial are reported in Table 1 and summarised in Figure 1. Overall, the mean speed across trials was 3.38 ± 0.7 m s^−1^ (mean ± 2 s.d.), including in-hand data, based on the available hoof sensor raw data. The speed the linear mixed models were evaluated ranged from 3.41–3.44 m s^−1^. The horses trotted faster with the jockey in the two-point seat position. The relationships between the different stride cycle parameters and speed are illustrated in Figure 2.

## 4. Discussion

This study demonstrated that surface type and jockey position can influence the hoof kinematics of trotting racehorses throughout a stride cycle.

Shorter hoof landing durations on tarmac suggest that the horses secured their footing more quickly on this firm and level surface than on the softer and more uneven artificial and grass surfaces. Importantly, if horses establish a secure hoof-ground contact rapidly, they can more safely and effectively adjust their movements as required. In the context of access routes to gallop tracks, this may minimise the number of negative interactions with other road or path users, thereby improving the health, well-being, and safety of these horses (and people). It is also relevant in the context of injury prevention. Previous work has identified that uneven racetrack surfaces place irregular vertical forces on the hooves of racehorses, which may influence soundness [31], and inconsistent arenas have been found to increase susceptibility to lameness in dressage horses [6]. Compared to the tarmac, the higher degree of surface roughness on the grass and artificial surfaces investigated in this study may have caused the horses’ hooves to be unstable for longer. These soft deformable surfaces were probably also associated with increased hoof slip and sink, essential for damping concussive forces at landing [20,21,22,23] but also likely to contribute to a prolonged landing phase.

Significantly different landing durations were apparent between both ridden positions on the tarmac and both ridden positions on the grass and artificial surfaces, thereby emphasising the dominant effect of surface on hoof landing duration. The most marked differences arose between the rising trot on the tarmac compared to the rising trot on the artificial surface (Δ66 ms), followed by the rising trot on the artificial compared to the two-point seat on the tarmac (Δ61 ms) (Appendix A). The soft artificial surface may have permitted greater hoof sink at landing than the perceivably firmer grass surface. This may explain why the largest difference arose between the tarmac and the artificial surface. Slightly longer landing durations on the artificial surface, relative to grass, may also be associated with more oblique forelimb landing and a higher range of motion of the hooves into this softer, more deformable surface, a factor that has been linked to longer forelimb landing durations on an artificial surface compared to grass in cantering horses [32].

The wide range in landing duration values on grass, compared to other surface conditions, involving several outliers (Table 1, Figure 2 and Figure 3), could reflect the greatest irregularity of this surface type [11] and perhaps the differing ease with which individual horses can respond and stabilise their hooves on unpredictable ground conditions. In contrast, the range in landing duration values on the firm and regular tarmac was tightly constrained (Figure 1). Alternatively, the lack of outliers in the data collected on the tarmac could reflect the optimisation of the hoof sensor algorithm for data collection on firm ground. 

This study observed a significantly shorter mid-stance duration on the artificial surface than on the tarmac. Previous work has suggested that surfaces that deform more increase stance time [33,34]. For example, stance duration was found to increase on deep wet sand (13.5% moisture) compared to firm wet sand (19% moisture) [33]. Maximal fetlock extension may also be reduced and delayed on deep wet sand or less stiff synthetic surfaces with greater damping properties [33,35], which may also prolong mid-stance. The data in our study were collected on a dry day, but it is plausible that mid-stance duration on the grass and artificial surface would vary with track moisture. Drawing direct comparisons between studies is also challenging, as landing and breakover phases may be incorporated into the definition of ‘stance’; for example, to delimit the stance phase, some previous studies have used a vertical force threshold of 50 N or 100 N, in each case stating this is from hoof first contact to toe-off [33,36,37]. Further work is needed to establish the role of surface moisture on the duration of the different stride phases. The artificial surface may compact with increased moisture, limiting hoof sink at landing. However, perhaps on grass, the surface would drain less well and remain more slippery, thereby prolonging the landing phase. Mid-stance duration had a moderate negative linear correlation with landing duration (r^2^ = 0.5, *p* < 0.001; Figure 3A). This relationship is logical as a shorter landing duration permits more time for the hoof to spend in the mid-stance phase within a given stride time.

Furthermore, the data show a separation according to surface type, with shorter landing and longer mid-stance for tarmac compared to longer landing and shorter mid-stance for grass and artificial surfaces. Increasing the duration of the mid-stance phase increases the time when the vertical force through the limb is high and, therefore, may contribute to stability. However, if a longer mid-stance phase is happening at the expense of a reduced landing time and slip period it may result in higher, potentially damaging, concussive forces when the body collides with the hoof [22,38,39,40]. There is a direct link between temporal data, such as stance time and ground reaction forces [41,42], but the nuances of surface and jockey position effects on the profile of the vertical ground reaction force magnitude through time would be interesting to evaluate in future; for example, using equipment that records the force and area loaded by hooves in motion [43].

Rider position is relevant at mid-stance. Although horses may modify their gait to compensate for changing surface properties [15,44], sudden alterations may challenge balance and increase loads on the musculoskeletal system [45]. In addition, the relatively passive properties of the equine distal limb [46] suggest that horses do not adjust limb stiffness when surface properties change [45], unlike humans, whose limb stiffness may adapt immediately to variations in surface stiffness [47]. Therefore, for soft and uneven surfaces, it may be helpful for a jockey to adopt a two-point seat position, which isolates their centre of mass from that of the horse [48] and reduces the amount of work the horse needs to do in accelerating their mass forwards and backwards per stride cycle, while also extending the period of weight-bearing at mid-stance (Table 3). Adaptations should also be made in the context of variable surface properties, such as moisture content, which control surface consistency [49].

For the rising trot data, it would be interesting to interpret the hoof kinematics in relation to the two halves of the stride cycle, depending on whether the jockey is seated in the saddle or standing in their stirrups. Although it is currently not possible to acquire this information from the inertial sensor technology fitted to the hooves, a complementary aspect to this study involved an assessment of upper body displacement patterns in the horses and jockey using different inertial sensors (XSens MTW) (Horan et al., in prep.). The latter sensors provide raw displacement data for all points of the stride cycle. Data from these sensors indicated that the magnitude of the horses’ vertical movement in their upper body and the time delay between horse and jockey cyclical movements over the stride cycle are sensitive to the differing position of the jockey in the diagonal stance phases of rising trot (Horan et al., in prep.).

Although the surface type and jockey position were not found to influence breakover in the ridden data sets, it was apparent that on tarmac breakover durations were considerably reduced, by approximately 70%, in the presence of a jockey, suggesting propulsion is sensitive to rider presence (Figure 1). Therefore, future studies assessing breakover should consider whether the horse is ridden or unridden. Perhaps the jockey has a role in encouraging a more rapid propulsive stage, or this effect may reflect that the horses trotted at lower speeds in-hand (Figure 1).

Swing duration increased on grass compared to the other surfaces, with a significant difference between grass and artificial of 20 ms. This could be related to a reduced duration of hoof sinking on grass at landing compared to on the artificial surface permitting more time for extension of the distal limb during swing. If the horses could extend their limbs further over a longer swing period on grass, this may contribute to the longest stride lengths on grass. It is counterintuitive that increased speed would be associated with an increased swing phase (Figure 2D), as the timings of the different stride phases would be expected to compress as the horses trot faster; a relationship between decreasing stride time with increasing speed is indeed true for galloping horses [50]. One possibility is that a faster trot provoked greater extension of the forelimbs and/or higher limb elevation, prolonging the swing phase over the narrow range of speeds studied. The other phases of the stride cycle may have been reduced sufficiently to compensate for this effect, particularly the mid-stance duration (Figure 2B). This study is also limited by its small sample size and the precision of the sensors, which to date have only been validated against motion capture data using strides from one horse [28]. The current algorithm used to integrate the acceleration data from the sensors may produce errors if certain assumptions are not met; for example, most of the outliers in the data appear to be associated with the soft surfaces (Figure 2 and Figure 3). In addition, the convenience basis on which the participants were recruited for this pilot study, involving retired racehorses and only one jockey, means the horses’ hoof kinematics reported may not accurately represent those of racehorses in active competitive work. For example, maturity and increasing stiffness of the suspensory apparatus tissues in older Thoroughbreds lessen the dorsi-flexion of forelimb fetlock joints [51,52].

Overall, stride durations were reduced on the tarmac. This result appears to be driven by the reduced landing durations on this surface (Figure 1). It does conflict with previous studies assessing stride times at trot on different surfaces [12,15]. However, this study differs from previous work where the horses were either driven [15] or ridden in a sitting trot [12], and it is worth noting that the horses trotted fastest with the jockey in the two-point seat position in this study. Previously, no difference in stride durations in a mixed population of horses ridden in sitting trot on tarmac, grass and artificial surfaces was observed [12], and the trot kinematics of harness trotters on asphalt versus sand indicated an increased stride frequency on the softer sand surface [15]. However, we did not find the stride durations of the horses to be affected by the jockey position. Stride lengths were also unaffected by jockey position, which appears to concur with the observation that stride lengths of horses on the approach to a jump and take-off and landing distances are not influenced by rider experience and associated differences in body position and movement [53]. However, it is worth emphasising that horses can alter stride duration and length characteristics differently with speed in response to environmental conditions [15] or before a musculoskeletal injury. For example, a recent study identified an association between a decrease in speed and stride length—but not stride frequency—and an increased risk of musculoskeletal injury in galloping racehorses, with stride characteristics changing markedly approximately six races before injury [54]. Our study also found a clear link between stride length and speed (Figure 2F), but not stride duration and speed. The potential impact of speed on the data emphasises the importance of including speed as a covariate in models assessing gait kinematics. It would be interesting to explore injury data for trotting racehorses in the context of these fundamental stride parameters. It would also be informative to determine how the stride timing parameters vary as horses move over inclined surfaces.

## 5. Conclusions

The duration of stages in a horse’s stride cycle and stride length during trot may be influenced by a combination of ridden seating style and/or ground surface properties. Using retired Thoroughbred racehorses, this study has demonstrated that longer hoof landing durations were associated with soft and irregular surfaces (grass and artificial). When landing times were lengthened, the subsequent mid-stance phase duration was typically reduced, but variation in landing duration appeared to be the main influence on total stride duration. Increasing the duration of mid-stance increases the time over which there is a large vertical ground reaction force on the distal limb, and this may contribute to a better constraint on limb positioning and therefore increased stability. Breakover was not significantly influenced by surface or jockey position in this study. Swing duration was longest on grass. Stride length was closely correlated to speed. From the perspective of hoof kinematics, although surface had the dominant effect, if small improvements in stability in the horse-jockey dyad are sought then this may be achieved by the jockey adopting a two-point seat position and thereby extending the mid-stance period. Future work will report the relationship between the hoof and upper body kinematics in the horse-jockey dyads under these conditions and determine the relative phasing of the horse and jockey displacements during the stance and flight phases of the stride cycle.

## Figures and Tables

**Figure 1 animals-13-02563-f001:**
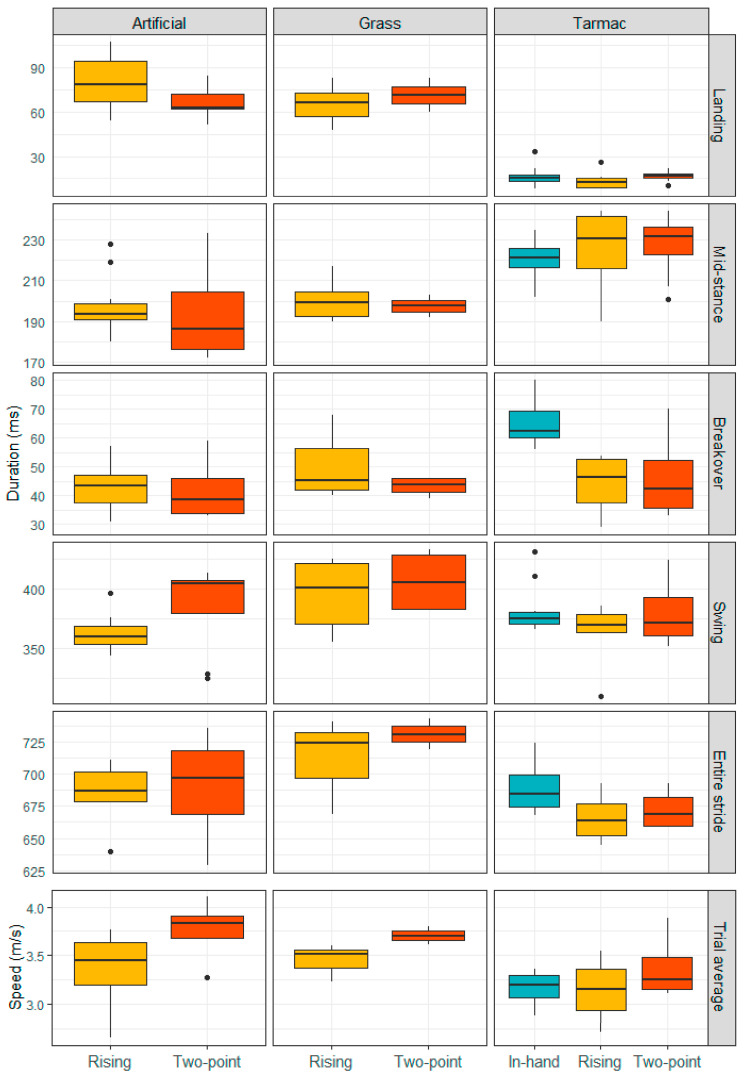
Boxplots displaying the spread in data for the median duration of landing, mid-stance, breakover, swing and total stride time, sub-divided according surface type and colored by jockey position. Distribution in speed across trials is also shown.

**Figure 2 animals-13-02563-f002:**
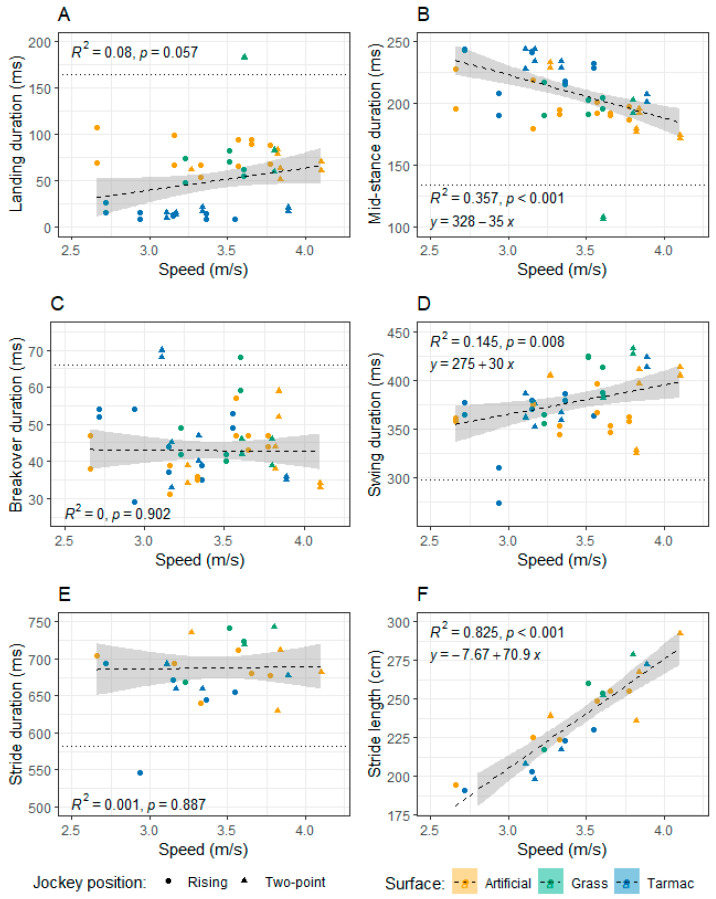
Relationship between stride parameters and speed on the artificial, grass and tarmac surfaces, with the jockey adopting rising and two-point seat positions in the ridden trials. Stride parameters are as follows: (**A**) Landing duration; (**B**) Mid-stance duration; (**C**) Breakover duration; (**D**) Swing duration; (**E**) Stride duration; and (**F**) Stride length. Thresholds for outliers, indicated by the dashed lines, were calculated based on ridden data. Points outside the dashed lines were excluded from the regression analysis and the linear mixed models.

**Figure 3 animals-13-02563-f003:**
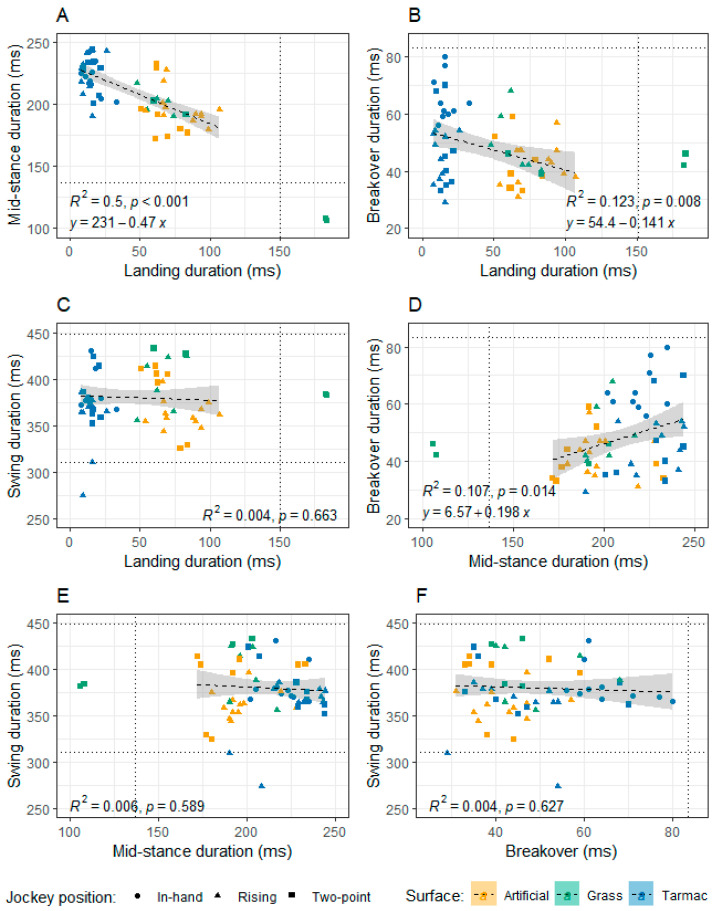
Relationships between the durations of the different stride cycle components on the artificial, grass and tarmac surfaces, with the horse trotted in-hand and ridden in rising and two-point seat positions. Stride cycle components are as follows: (**A**) Mid-stance duration versus landing duration; (**B**) Breakover duration versus landing duration; (**C**) Swing duration versus landing duration; (**D**) Breakover duration versus mid-stance duration; (**E**) Swing duration versus mid-stance duration; and (**F**) Swing duration versus breakover duration. The dashed lines represent the boundaries for values 1.5 times the inter-quartile range below quartile 1 or 1.5 times the inter-quartile range above quartile 3. Data points were deemed outliers if they fell outside the dashed lines, and these were not included in the linear regression analyses indicated.

**Table 1 animals-13-02563-t001:** Hoof kinematic variables that were recorded by a hoof-mounted inertial sensor system during ridden trials.

Horse	Year Born	Height (hh)	Mass (kg)	Jockey position	Surface	Limb	No. of Strides	Trot Speed (m s^−1^)	Median Stride Length (m)	Mid-Stance (s)	Error Mid-Stance +/− (s) *	Median Breakover Duration (s)	Error Breakover +/− (s)	Swing (s)	Swing Error +/− (s)	Median Landing Duration (s)	Landing Error +/− (s)	Stride Duration (s)	Relative Stance Time (%)
1	2004	16.1	550	Rising	Artificial	LF	47	3.65	2.55	0.192	0.023	0.043	0.007	0.354	0.039	0.09	0.04	0.68	47.8
				Rising	Artificial	RF	47	3.65	2.55	0.19	0.036	0.047	0.008	0.347	0.032	0.094	0.038	0.68	49
				Two-point	Artificial	LF	14	3.82	2.36	0.177	0.058	0.038	0.009	0.329	0.058	0.084	0.075	0.629	47.7
				Two-point	Artificial	RF	14	3.82	2.36	0.18	0.038	0.044	0.007	0.325	0.043	0.079	0.042	0.629	48.3
2	2007	16	520	Rising	Tarmac	LF	5	2.94	1.64 †	0.19	0.019	0.029	0.01	0.31	0.05	0.016	0.007	0.547 ‡	43.3
				Rising	Tarmac	RF	5	2.94	1.64 †	0.208	0.016	0.054	0.004	0.274 ‡	0.059	0.009	0.001	0.547 ‡	49.8
				Two-point	Tarmac	LF	58	3.89	2.72	0.207	0.024	0.036	0.01	0.414	0.021	0.021	0.012	0.678	39
				Two-point	Tarmac	RF	58	3.89	2.72	0.201	0.013	0.035	0.008	0.424	0.031	0.017	0.013	0.678	37.5
				Rising	Artificial	LF	26	2.66	1.94	0.196	0.055	0.038	0.016	0.362	0.063	0.107	0.051	0.704	48.5
				Rising	Artificial	RF	27	2.66	1.94	0.228	0.053	0.047	0.01	0.358	0.086	0.069	0.063	0.704	49.1
				Two-point	Artificial	LF	10	3.27	2.39	0.233	0.023	0.034	0.007	0.406	0.021	0.062	0.04	0.736	44.9
				Two-point	Artificial	RF	11	3.27	2.39	0.229	0.037	0.039	0.009	0.405	0.021	0.062	0.022	0.736	44.9
				Rising	Grass	LF	54	3.51	2.6	0.191	0.072	0.04	0.009	0.425	0.028	0.083	0.074	0.741	42.6
				Rising	Grass	RF	56	3.51	2.6	0.203	0.041	0.042	0.009	0.424	0.04	0.07	0.045	0.741	42.7
				Two-point	Grass	LF	45	3.8	2.79	0.192	0.053	0.039	0.006	0.427	0.02	0.083	0.054	0.743	42.5
				Two-point	Grass	RF	46	3.8	2.79	0.203	0.026	0.046	0.008	0.433	0.024	0.06	0.036	0.743	41.7
3	2014	16	530	Rising	Tarmac	LF	42	3.55	2.3	0.232	0.025	0.049	0.012	0.364	0.026	0.009	0.006	0.655	44.5
				Rising	Tarmac	RF	42	3.55	2.3	0.229	0.022	0.053	0.009	0.364	0.016	0.008	0.006	0.655	44.4
				Two-point	Tarmac	LF	18	3.11	2.08	0.244	0.008	0.070 ‡	0.006	0.362	0.015	0.016	0.009	0.693	47.8
				Two-point	Tarmac	RF	18	3.11	2.08	0.228	0.014	0.068 ‡	0.01	0.386	0.014	0.01	0.005	0.693	44.3
				Rising	Artificial	LF	43	3.77	2.55	0.198	0.031	0.047	0.007	0.363	0.033	0.068	0.048	0.678	46.5
				Rising	Artificial	RF	42	3.77	2.55	0.187	0.075	0.044	0.01	0.358	0.029	0.088	0.078	0.678	47.2
				Two-point	Grass	LF	9	3.61	2.52	0.108 ‡	0.032	0.042	0.007	0.384	0.013	0.183 ‡	0.029	0.719	46.6
				Two-point	Grass	RF	7	3.61	2.52	0.106 ‡	0.043	0.046	0.004	0.382	0.013	0.184 ‡	0.035	0.719	46.9
4	2012	17	580	Rising	Tarmac	LF	10	2.72	1.91	0.244	0.031	0.052	0.015	0.377	0.047	0.016	0.017	0.693	46
				Rising	Tarmac	RF	9	2.72	1.91	0.243	0.037	0.054	0.015	0.365	0.029	0.026	0.034	0.693	47
				Two-point	Tarmac	LF	13	3.17	1.98	0.234	0.018	0.033	0.005	0.376	0.04	0.013	0.003	0.659	42.8
				Two-point	Tarmac	RF	12	3.17	1.98	0.244	0.018	0.045	0.011	0.352	0.017	0.016	0.007	0.659	46.5
				Rising	Artificial	LF	55	3.57	2.49	0.201	0.032	0.047	0.013	0.397	0.029	0.066	0.048	0.711	44.2
				Rising	Artificial	RF	52	3.57	2.49	0.192	0.045	0.057	0.009	0.367	0.05	0.094	0.06	0.711	48.4
				Two-point	Artificial	LF	26	3.84	2.67	0.196	0.039	0.052	0.007	0.411	0.025	0.051	0.039	0.712	42.3
				Two-point	Artificial	RF	26	3.84	2.67	0.192	0.039	0.059	0.011	0.396	0.034	0.063	0.043	0.712	44.3
				Rising	Grass	LF	49	3.6	2.54	0.196	0.044	0.059	0.026	0.414	0.036	0.055	0.047	0.724	42.9
				Rising	Grass	RF	49	3.6	2.54	0.205	0.031	0.068 ‡	0.013	0.388	0.035	0.062	0.038	0.724	46.4
5	2007	16	510	Rising	Tarmac	LF	7	3.36	2.23	0.218	0.01	0.035	0.006	0.386	0.006	0.008	0.002	0.645	40.3
				Rising	Tarmac	RF	9	3.36	2.23	0.215	0.022	0.039	0.007	0.379	0.015	0.014	0.015	0.645	41.4
				Rising	Artificial	LF	11	3.16	2.25	0.219	0.016	0.031	0.006	0.376	0.03	0.067	0.039	0.694	47.2
				Rising	Artificial	RF	9	3.16	2.25	0.18	0.042	0.039	0.007	0.375	0.027	0.099	0.054	0.694	46.1
				Two-point	Artificial	LF	8	4.1	2.92	0.172	0.022	0.034	0.006	0.414	0.03	0.061	0.029	0.682	39.1
				Two-point	Artificial	RF	6	4.1	2.92	0.174	0.018	0.033	0.005	0.405	0.025	0.07	0.025	0.682	39.8
6	2002	16.3	550	Rising	Artificial	LF	50	3.33	2.24	0.191	0.058	0.036	0.01	0.344	0.051	0.067	0.062	0.64	46.2
				Rising	Artificial	RF	53	3.33	2.24	0.195	0.035	0.035	0.007	0.354	0.045	0.054	0.039	0.64	44.6
				Rising	Tarmac	LF	11	3.15	2.03	0.242	0.014	0.044	0.007	0.37	0.025	0.013	0.007	0.672	44.7
				Rising	Tarmac	RF	11	3.15	2.03	0.241	0.015	0.037	0.005	0.379	0.011	0.012	0.006	0.672	43.2
				Two-point	Tarmac	LF	12	3.34	2.17	0.229	0.039	0.047	0.009	0.359	0.022	0.022	0.024	0.66	45.4
				Two-point	Tarmac	RF	12	3.34	2.17	0.234	0.022	0.04	0.01	0.367	0.028	0.017	0.009	0.66	43.4
				Rising	Grass	LF	7	3.23	2.17	0.217	0.021	0.049	0.007	0.356	0.03	0.048	0.03	0.669	46.5
				Rising	Grass	RF	7	3.23	2.17	0.19	0.058	0.042	0.009	0.365	0.032	0.074	0.059	0.669	45.2

* Error indicates where approximately 90% of the values lie within. ‡ These values were deemed outliers and excluded from the Linear Mixed Models. † This value reported by the sensor algorithm was excluded from Linear Mixed Model analyses as it is incorrect (it is the same as the value indicated for walk in the same trial).

**Table 2 animals-13-02563-t002:** Significance values from Linear Mixed Models (final models).

Parameter	Source	F Value	Significance
Landing duration (ms)	Jockey position	0.23	0.636
	Surface	100.88	<0.001
	Speed	0.17	0.679
	Jockey position*Surface	3.71	0.035
Mid-stance duration (ms)	Jockey position	13.34	<0.001
	Surface	8.25	0.001
	Speed	30.71	<0.001
	Jockey position*Surface	1.40	0.258
	Jockey position*Speed	12.22	0.001
Breakover duration (ms)	Jockey position	3.34	0.076
	Surface	0.72	0.495
	Speed	0.35	0.556
Swing duration (ms)	Jockey position	1.26	0.268
	Surface	3.54	0.039
	Speed	12.34	0.001
Stride duration (ms)	Jockey position	0.62	0.446
	Surface	6.30	0.011
	Speed	0.93	0.351
Relative stance time (%)	Jockey position	0.16	0.688
	Surface	14.74	<0.001
	Speed	51.45	<0.001
Stride length (cm)	Jockey position	0.24	0.63
	Surface	4.25	0.036
	Speed	133.28	<0.001

**Table 3 animals-13-02563-t003:** Estimated Marginal Means for jockey position effects.

Parameter	Jockey Position	Mean	Std. Error	df	95% Confidence Interval (Lower Bound)	95% Confidence Interval (Upper Bound)
Landing duration (ms)	Two-point	51.6	4.3	12.0	42.2	61.0
	Rising	53.7	3.1	4.7	45.5	61.9
Mid-stance duration	Two-point	219.7	4.3	38.0	211.0	228.5
	Rising	206.7	2.6	38.0	201.5	211.9
Breakover duration (ms)	Two-point	40.9	2.7	9.4	34.7	47.0
	Rising	44.8	2.5	6.7	38.9	50.7
Swing duration (ms)	Two-point	380.2	10.0	6.3	356.2	404.3
	Rising	373.5	9.5	5.3	349.5	397.5
Stride duration (ms)	Rising	684.6	9.9	7.3	661.3	707.8
	Two-point	692.8	10.9	9.5	668.3	717.2
Relative stance (%)	Two-point	45.0	1.0	6.1	42.7	47.4
	Rising	45.2	0.9	5.3	42.9	47.6
Stride length (cm)	Rising	237.0	3.9	6.9	227.7	246.4
	Two-point	235.1	4.3	8.9	225.4	244.8

**Table 4 animals-13-02563-t004:** Estimated Marginal Means for surface effects.

Parameter	Surface	Mean	Std. Error	df	95% Confidence Interval (Lower Bound)	95% Confidence Interval (Upper Bound)
Landing duration (ms)	Grass	68.9	5.6	20.3	57.3	80.6
	Tarmac	16.0	3.5	7.3	7.7	24.3
	Artificial	72.9	3.4	6.8	64.7	81.0
Mid-stance duration (ms)	Grass	210.3	5.5	38.0	199.2	221.4
	Tarmac	224.1	3.2	38.0	217.7	230.4
	Artificial	205.3	3.4	38.0	198.5	212.1
Breakover duration (ms)	Grass	44.7	3.1	13.7	38.1	51.3
	Tarmac	42.1	2.7	8.6	36.1	48.2
	Artificial	41.7	2.5	7.5	35.8	47.6
Swing duration (ms)	Grass	388.5	10.9	8.7	363.8	413.2
	Tarmac	373.2	10.0	6.3	349.1	397.2
	Artificial	368.9	9.7	5.8	344.9	392.9
Stride duration (ms)	Grass	713.8	13.1	13.9	685.7	741.9
	Tarmac	665.1	11.2	10.3	640.4	689.8
	Artificial	687.1	10.3	8.5	663.6	710.7
Relative stance time (%)	Grass	45.6	1.0	8.0	43.2	48.0
	Tarmac	43.4	1.0	6.0	41.0	45.8
	Artificial	46.4	0.9	5.6	44.1	48.8
Stride length (cm)	Grass	242.3	5.1	13.3	231.3	253.2
	Tarmac	228.2	4.4	9.7	218.4	238.0
	Artificial	237.8	4.1	8.0	228.3	247.2

**Table 5 animals-13-02563-t005:** Estimated Marginal Means for jockey position and surface effects.

Parameter	Jockey Position	Surface	Mean	Std. Error	df	95% Confidence Interval (Lower Bound)	95% Confidence Interval (Upper Bound)
Landing duration (ms)	Two-point	Grass	70.8	9.0	36.6	52.6	89.0
		Tarmac	18.2	4.6	16.7	8.6	27.8
		Artificial	65.6	5.0	21.0	55.3	75.9
	Rising	Grass	67.0	5.2	21.2	56.2	77.8
		Tarmac	13.9	4.4	16.4	4.6	23.1
		Artificial	80.1	3.8	11.1	71.7	88.5
Mid-stance duration (ms)	Two-point	Grass	219.8	9.7	38.0	200.2	239.4
		Tarmac	225.5	4.4	38.0	216.5	234.5
		Artificial	213.9	5.7	38.0	202.5	225.4
	Rising	Grass	200.8	5.1	38.0	190.4	211.1
		Tarmac	222.7	4.5	38.0	213.6	231.7
		Artificial	196.7	3.6	38.0	189.3	204.0
Breakover duration (ms)	Two-point	Grass	42.7	3.4	18.3	35.6	49.9
		Tarmac	40.1	2.9	11.5	33.8	46.5
		Artificial	39.7	2.9	12.1	33.4	46.1
	Rising	Grass	46.7	3.1	14.4	40.0	53.3
		Tarmac	44.1	2.8	11.0	37.9	50.3
		Artificial	43.7	2.6	8.2	37.7	49.7
Swing duration (ms)	Two-point	Grass	391.9	11.7	11.3	366.3	417.5
		Tarmac	376.5	10.3	7.1	352.3	400.7
		Artificial	372.3	10.4	7.7	348.0	396.5
	Rising	Grass	385.2	10.9	8.8	360.5	409.8
		Tarmac	369.8	10.5	7.8	345.5	394.1
		Artificial	365.5	9.9	6.1	341.5	389.5
Relative stance time (%)	Two-point	Grass	45.5	1.1	9.9	43.0	48.0
		Tarmac	43.3	1.0	6.7	40.9	45.7
		Artificial	46.3	1.0	7.1	43.9	48.7
	Rising	Grass	45.7	1.0	7.9	43.3	48.1
		Tarmac	43.5	1.0	7.0	41.1	45.9
		Artificial	46.5	1.0	5.9	44.2	48.9
Stride duration (ms)	Rising	Grass	709.7	13.2	14.5	681.6	737.9
		Tarmac	661.0	12.8	14.2	633.6	688.4
		Artificial	683.0	10.8	9.9	658.9	707.2
	Two-point	Grass	717.9	15.0	16.4	686.2	749.6
		Tarmac	669.2	11.8	12.1	643.4	695.0
		Artificial	691.2	12.3	13.4	664.8	717.7
Stride length (cm)	Rising	Grass	243.2	5.1	13.7	232.3	254.2
		Tarmac	229.2	5.0	13.3	218.5	239.9
		Artificial	238.7	4.3	9.1	229.1	248.3
	Two-point	Grass	241.3	5.8	15.9	229.1	253.6
		Tarmac	227.3	4.6	11.2	217.1	237.4
		Artificial	236.8	4.8	12.4	226.4	247.2

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
