# Peer review of "Timing Differences in Stride Cycle Phases in Retired Racehorses Ridden in Rising and Two-Point Seat Positions at Trot on Turf, Artificial and Tarmac Surfaces"

_animals, 2023, doi:10.3390/ani13162563_

Round 1

Reviewer 1 Report

With interest, I have been reading the manuscript entitled: "Hoof kinematics of Thoroughbreds ridden in rising and two-point seat positions at trot on turf, artificial and tarmac surfaces" by Horan et al.

Congratulations on the manuscript. I have imagined how much hard work was involved, and it may have contributed to other researchers' training within the research group. The outcome is clear. I have a few comments and questions that I have detailed below.

 Suggestions for improving the manuscript:

 Abstract:

L56-57:

My comments: I wonder if this phrase has a relationship between the title and the goals of this study.

Introduction

My comments: Well-written and straightforward.

Materials and Methods

Was this study made from a cross-over design? The complete experimental design must make part of the MM in detail. Was there recovery period (washout) among trials?

L146: Convenience sampling results cannot be generalized to the target population due to potential sampling technique bias due to the underrepresentation of groups in the sample compared to the population of interest. Sample bias cannot be measured. Inferences based on convenience sampling should only be made about the sample itself.

Please, clarify it and elaborate more in the manuscript bringing the advantages and disadvantages of applying with a convenience sample. This limitation should be included when discussing and reviewing the results of the current study.

 L190-191: Do you have previous literature on this statement?  Please, provide a reference for this methodology.

 Results

Table 1: The table formatting needs to be improved. Wouldn't it be better to put this table as a supplementary file? 

Discussion

As previously mentioned, the authors need to bring a study limitation on the convenience sample.

Reviewer 2 Report

While the study of riding surfaces has drawn research interest over the years within the equine industry, this field of study still has work that needs to be done and is relevant in particular to the racehorse industry. As such, the topic area for this manuscript holds value to the industry. However, there are several significant limitations to the methodology of this study that causes this work to fall short of being able to offer any conclusive conclusions that aren't associated with further questions. First off, the use of IMU sensors, while practical for the industry, is not the accepted technology for scientific biomechanical research. Although if the authors were exploring the practicality of IMU sensors for racehorse training, this approach may hold some value. IMU sensors are truly limited to just temporal variable data in which more impactful data collecting specifically kinetic data, ground reaction forces, or further joint moments and powers may demonstrate the potential risk associated with different ground surfaces specific to the racehorse industry. The title should at least reflect that this research is specifically looking at temporal variables, remove hoof kinematics, and potentially adding in that this is IMU sensor data. Further, temporal variables are limited in their application and with the numerous studies that have been done several decades ago on the trot and the influence of riders and other variables on similar measurements done within this study this approach of only collecting temporal variable data is not offering anything new to the field of research as it currently stands.

Another limitation to this study is the use of geriatric, retired horses for the sample population. Does this age group reflect the population represented on the racetrack, and even if the title and introduction were changed to reflect the population utilized, what is the importance of measuring this older population as they are retired from the sport? It is interesting, however, to note that the title actually does not say anything about racehorses, although the introduction and discussion including the conclusions focuses solely on the racehorse industry. The title as such is quite misleading as to the focus of this work. Further, the objectives even mention "jockey", and yet, that is not given in the title. As such, the authors may want to adjust so the title matches that of the bulk of the manuscript when it comes to topic focus, but of course, does the bulk of the manuscript actually reflect the population utilized for the data collection? Further, especially with one horse over 20 years old and retired from the racetrack, these horses needed to be thoroughly evaluated prior to the study by a veterinarian, and since degenerative changes within the joints are not uncommon within the race industry and is a common reason for retirement, this evaluation needed to include diagnostic imaging to objectively check for these changes. Keep in mind that age does influence gait kinematics even for sound horses, and thus, without a larger more diverse population when it comes to ages, this study is limited to an older age group. As for soundness, was even gait symmetry verified by left-right measures? Do keep in mind, however, that asymmetry of locomotion can be induced by the diagonal selected by the rider for the rising trot due to uneven biphasic loading, and thus, determining symmetry of the gait by the left-right variables would need to keep in mind what diagonal was being utilized by the rider. From table 1 there is no mention of separation of variables specific to the diagonal utilized by the rider. Further, there is no mention as to how the gait was guaranteed that it was being performed correctly or as to whether the rider was performing the rising and/or two-point trots in a manner reflective of the industry. With such a small sample size to make a generalization that these horses and their respective performance within this study are standard for the industry it is important to ensure that these standards were met. Further, with only one rider, that standard becomes more critical as to whether this individual performed the rising trot and the two-point as an ideal representation of these gaits and of all jockeys performing these two riding styles. Multiple studies have demonstrated influence of rider, whether experienced or not, and thus, could these variables be more reflective of the rider than the ground surface and/or riding style being evaluated? As such, what was the inclusion/exclusion criteria utilized for strides that were included within the analysis? Were additional strides not reported, and if so, why? 

Research needs to be repeatable but the methods section leaves out much information concerning specifics associated with data collection. Take for example the following: what are the specifics to the surfaces utilize (i.e. what type of grass, what type of artificial surface, etc.), what was the specific location for data collection and was it the same for all, was each horse tracked over the same area for each surface and was it a level surface for all, what was the order of surfaces and riding styles during data collection, and was collection for all surfaces and riding styles for all horses done on the exact same day and if so how was fatigue of rider and horse ensured (i.e. was the horse/rider required to do consistently without breaks all of their 198 strides collected for the left variables for horse 2)? Since this study is evaluating various surfaces, more thorough details giving more objective measures concerning the type of surfaces needs to be given. Also, since speed wasn't controlled for, how was consistency of gait ensured between strides and between horses? As demonstrated by the figures and previous studies, speed influences all the variables measured in this study so that changes associated with inconsistencies of speed between strides due to fatigue for example could be an influencing factor more so than riding style or surface. Was there a control utilized for this study? Assuming the in-hand measures were considered a control but data concerning these measures is limited. Further, more details concerning race experience, specifically years racing for the horses is needed. In addition, as mentioned previously, the diagonal utilized for the rising trot can influence kinematics, and thus, was diagonal selection randomized and was data dissected to look at the influence of each diagonal? Further, was the riding style and surface type randomized to the order that they were performed? 

Along with the population not being reflective of those horses racing currently due to age and retirement status, the sample population is quite small with only 6 horses. Further, not all horses had data collected for all riding styles and/or surface types. In fact, it was only one horse that this objective was accomplished in which that horse for the rising trot on the tarmac surface only had five strides collected for each forelimb. Further, for grass surface type, only four horses performed over this surface and only 3 at the rising trot and only 2 at the two-point. This sample size would reflect more numbers for a pilot study.

As for the results and discussion, within table 1 there are additional variables that aren't discussed throughout the manuscript including hoof angle, maximum lateral movement and maximum hoof height. If not discussed, remove from table. Also, for those variables discussed, within the methods give clear definitions of each including specifically how it was measured. This is given for some variables, but not all. For reporting these variables, divide not between each variable within the results section and the discussion section, but rather between each riding style and each surface, i.e. the subsections for the results and discussion, so that the reader can understand the interrelationship between the variables and how they impact each other. This goes back to how speed selection alone, independent from riding style and surface, has a direct impact on all of the measurements collected within this study. Even the authors express this point within their figures, but  needs to be expanded within other variables and needs the data discussed accordingly including within the discussion section. For example, authors need to expand on how breakover impacts stance and in turn influences stride duration and how these variables make up the gait that is performed specific to the riding style and/or surface. None of these variables are independent of each other, and thus, need to be reported and discussed accordingly. Also, it is surprising that while stride length is given why wasn't stride frequency reported? This is a simple calculation from the data collected and can give further insight as to the dependency of stride length versus stride frequency in producing a faster gait.

The data collected, i.e. the temporal variables, as a whole is quite basic, and thus, doesn't require the number of tables and figures given within the manuscript, and in fact, seems redundant at times. Reduce accordingly so that the findings are streamlined and focused on the objectives. Table 1 is useful within the reporting, but ensure that the rows are consistently organized between horses as to the order the riding styles and surfaces are listed so that it is easy to compare between each horse. Finally, because of the simplicity of the data collected, the discussion section is too long. Again, organizing the discussion so that each variable isn't discussed within one paragraph by itself. In fact, a paragraph focused on riding style and then one focused on surface types would be sufficient. While the current discussion needs to be streamlined as to it's focus on what was uncovered so that it is not as lengthy, additional discussion must be added concerning study limitations. This review only touches on a few of these limitations that must be addressed. Finally, the conclusion also must be shortened and focusing solely on the results specific to comparison of riding style and to surface type avoiding predictions not validated by data. 

Formatting for in text citations was inconsistent, thus, correct accordingly. 

Reviewer 3 Report

Over all the study was well written, but not enough information was included in the methodology.  Also, please consider the data presented by Logan et al., on the accuracy of some tech in its use in horses for gait analysis.

Line 51 – extra commas

How were the horse’s shod?  Describe more precisely the placement of the IMU, how were they attached?  Have you done any testing on instruments to determine if accuracy decays with time?  How long after placement did the trials take place?  Did all trials take place on the same day?  Was it randomized which surface was tested first, as well as jockey position?  Were values for both limbs averaged?  Analyzed separately for difference between right and left limbs?

Could you comment on the shorter stride duration on tarmac?  Do the authors believe the horse was reducing stride to perhaps a greater insecurity on tarmac and more “careful” hoof placement?  Tarmac is typically associated with horses slipping, were you able to capture any data on instability?

Round 2

Reviewer 2 Report

Without redoing the research within this study, it is hard to address all of the concerns of the original reviews, and thus, authors are commended on trying to address what issues they can without discarding their current work. Nevertheless, the issues brought up in the original reviews still are a limitation, particularly the use of geriatric, retired racehorses. First, the interest of observing the performance of retired racehorses within a race environment does not appear to have much application beyond a few places that may utilize such animals for teaching purposes, and even then, this application seems quite small. The revised title of the manuscript is acceptable, but only points to this use of a population with limited relevance within the industry. This population may be more of value if the focus was on more clinical based research for potential treatment and management options of lameness than for performance application. The introduction is focused on active racehorses, not retired, and their daily activities in which this discussion within the introduction does not apply to whole of the retired population. Further, while supplementary data was provided concerning symmetry on hard surfaces, without a more thorough clinical assessment utilizing common, accepted diagnostic imaging tools, it is hard to guarantee that the health of the geriatric animal did not impact data. In addition, this reviewer is in agreement that the popularity of IMU data is growing, especially within a teaching environment, but as the authors point out this work within the scientific community has been "gradually increasing" within research "since the 2000s" making it a relatively new science within the field of equine research. As such, data supporting IMU sensor work from more well established methods is often advised. This is where the symmetry data presented verifying the soundness of the geriatric animals would be better supported by more tested methods of diagnostic imaging. Further, while research may not be specifically labeled for Thoroughbreds, there is substantial work done on dressage horses with several of these studies including Thoroughbred horses. These studies have included mostly the trot and specifically temporal variable measurements, and as such, may be useful for further exploring background research and determining the novelty of the current study. Similarly, surface studies have been available, even if again not specifically utilizing retired racehorses, and the influence of performance type outside of the racehorse has been studied. Nevertheless, the authors are correct that this study is unique, but as to it's application within the whole of the equine industry, that is unsure. In any case, a more thorough pubmed search may help to see studies where overlap is present and applicable previous data is available determining the relevance and novelty of this research in it's current state. Further, while authors shouldn't find any research on retired racehorses, this mays suggest the lack of need for this type of research. Potentially a more convincing introduction targeting this specific population with a more applicable objective statement and hypothesis statement may give a better understanding as to why retired racehorses need to be researched outside of clinical treatment and management. Finally, the sample size is quite small and while statistical analysis by the authors has provided limited confidence as to potential strength, the other limitations of this work pointed out in the previous reviews, only emphasizes the need of a larger sample size. The number of variables investigated should suggest the need for additional samples and with inclusion/exclusion criteria for this study not appearing to be too restrictive it is surprising that more participants couldn't have been recruited. Even the use of the IMU sensors would allow for ease of data collection and potentially even allowing for data collection of active racehorses to support application to that of retired racehorses. 

See current and previous reviews.